# Parent–Adolescent Conflict: Adolescents’ Coping Strategies and Preferred Parenting Styles during the Social Movements in Hong Kong, 2019

**DOI:** 10.3390/bs13090756

**Published:** 2023-09-12

**Authors:** Andrew Yiu Tsang Low

**Affiliations:** Felizberta Lo Padilla Tong School of Social Sciences, Caritas Institute of Higher Education, Hong Kong; alow@cihe.edu.hk

**Keywords:** social movements, parent-adolescent conflicts, political differences, parenting style, conflict resolution strategies

## Abstract

This research investigated parent–adolescent conflict, conflict resolution strategies and perceived parenting styles by adolescents during the social movements in Hong Kong in 2019, a period characterized by considerable social unrest in which many young people participated in demonstrations and protests. The study drew on responses from 866 adolescents aged between 11 and 16 who completed a questionnaire that included a conflict issue checklist and elicited respondents’ conflict resolution strategies as well as perceived parenting styles. Correlation analysis was performed to identify the correlation of parent–adolescent conflicts with differences in political stances with their parents and other demographic data. Regression analysis was performed to identify the correlation of perceived parenting styles and conflict resolution strategies adopted by adolescents. Results indicated that early adolescents have a higher intensity of conflicts with their parents than late adolescents in this period. Respondents had more intense conflicts with their parents over political differences and ways of expressing their political views than other issues. Those respondents in conflict or ineffective arguing strategies perceived their parents as more authoritarian than those who adopt positive conflict resolution strategies. However, when asked about their ideal ways of resolving conflicts, adolescents preferred problem-solving rather than conflict strategies.

## 1. Introduction

Social unrest in 2019 in Hong Kong, characterized by many demonstrations, the obstruction of roads, and vandalism, was triggered by the government’s extradition bill, revealing its underlying subjugation to the Chinese government, economic pressures resulting from high housing costs, and the lack of upward mobility opportunities for young people [1]. Many families experienced conflict between family members because of polarized political views, generating much stress in families with an adolescent child [2]. Adolescents’ political stances and how they choose to express their political views may differ from those of their parents. Parents worried about the safety of their children on the street participating in demonstrations, “Lennon walls” (walls covered with notes containing message and calls for democracy), obstructing roads, and forming human chains (protests to show the pro-democracy movement), even if they supported their views. Some parents who disagree with their adolescent children’s political views and actions may even prohibit them from coming home. Tensions in families, especially parent–adolescent conflict, place an extra burden on the already-tense relationship in response to adolescent development. Many parents feel frustrated by disagreements with their adolescent children, while at the same time some adolescents consider their parents as not accepting or understanding their thoughts.

### 1.1. Nature and Extent of Parent–Adolescent Conflict

Parent–adolescent conflict has been defined as “disagreement and opposition behavior between parents and their teenage children, together with the expression of negative emotional response between parents and their teenage children” [3]. Social cognitive theory [4] views self-efficacy as an important component of adolescence which adolescents need in order to face challenges in this period. Adolescents with high self-efficacy cope better than those with low self-efficacy. Parents with warm, responsive, and supportive parenting attitudes are more likely to foster high self-efficacy in their adolescent children [5]. Previous studies on Hong Kong have indicated that parents and their adolescent children engage in major conflicts over everyday issues. Hong Kong adolescents have also reported conflicts with parents because of smoking and their choices of leisure activities [6]. Nevertheless, previous research has not investigated disagreement on political views, such as that which occurred during the social movements in 2019. From a developmental perspective, parent–adolescent conflicts are normative during early adolescence, when adolescents strive for autonomy and the capacity to make more decisions by themselves. Therefore, parent–adolescent conflict is a necessary process enabling adolescents to renegotiate boundaries and their roles in their families, as well as ensuring the development of autonomy and identity, which are essential for separating them from their families [7]. Whether parent–adolescent conflict is good or bad for adolescents depends on the frequency of conflicts and how they cope with conflicts [8]. Nevertheless, research has identified intense parent–adolescent conflict as a major cause of negative family functioning [9]. Research indicates that low levels of family warmth and intense levels of parent–adolescent conflict correlate with adolescents’ suicidal ideation and depression in Hong Kong [10]. In addition, they correlate with delinquency [11] and running away from home [12]. Higher conflict levels are also associated with lower levels of psychological well-being and school adjustment, lower levels of adolescent self-satisfaction [13], and a higher level of substance use. 

### 1.2. Moderating Role of Conflict Resolution Strategies and Parent–Adolescent Conflict

Coping is a strategy for managing stress. Kurdek identified four categories of adolescents’ coping strategies: “conflict engagement”, in which verbally abusive, angry, and defensive behaviors, attacks, and displays of anger are observed; “withdrawal”, which involves avoiding the problem, avoiding talking, and becoming distant; “compliance”, accepting the antagonist’s perspective; and “positive problem-solving”, understanding the antagonist’s position and working out a compromise [14]. Research indicates that adolescents who adopt positive problem-solving and conflict resolution strategies can moderate the effect of parent–adolescent conflict. These conflict resolution strategies are also essential to maintaining good relationships between the respective actors [15]. Moreover, high levels of parent–adolescent conflict negatively correlate with effective conflict resolution and adolescent depression [16]. Conflict engagement or avoidance have been identified as being related to increased delinquency [17] and externalizing problems, like fighting, smoking, alcohol consumption, and marijuana use [18]. The adoption of positive conflict resolution strategies also predicts life satisfaction among adolescents. In the present study, adolescents’ adoption of different conflict coping styles during episodes of parent–adolescent conflict was explored within the framework of Kurdek’s conflict resolution strategies [14]. The relationship between different coping strategies and adolescents’ behaviors was also examined. It was hypothesized that adolescents who adopted negotiation and conflict resolution strategies would engage in fewer and less intense conflicts with their parents.

### 1.3. Parenting Style and Parent–Adolescent Conflict

Parents who are less responsive to their adolescent children’s needs and adopt controlling parenting behaviors are more likely to experience high levels of conflict with their children [19]. An authoritarian parenting style, characterized by low responsiveness, high demandingness, and harsh discipline, is positively associated with lower cohesion between parents and their adolescent children, higher conflict frequency [20], and more total conflicts. A lack of affection towards their children may also be related to depression and anxiety among their children and adolescents [21]. In addition, chronic delinquents’ home lives are characterized by their parents’ inability to instill proper standards and behavior related to self-control and hostility in parent–child conflicts. Parental rules that only limit children’s freedom my affect children’s relationship with authority [22]. Conversely, parents who demonstrate high levels of affection, nurturance, and empathy towards their children, as well as adopting an authoritative parenting style, reported less parent–adolescent conflict. Parents who have high expectations of their adolescent children’s behaviors, while providing affection and support and communicating with them, are less likely to experience high levels of conflict with their adolescent children, and their children are less likely to engage in antisocial behavior and experience lower life satisfaction. They demonstrate more cohesion with their children and encounter less frequent and less intense conflicts. The behavioral family system [23] model provides a theoretical framework for understanding parent–adolescent conflicts and was adopted in this study. Within this framework, developmental factors, a lack of problem-solving skills, inappropriate functional interactions (e.g., parenting style and interactions among family members), unresolved differences in political views, and an unsupportive family structure lead to parent–adolescent conflicts. The ability of individuals and families to apply problem-solving skills is one of the most important factors determining the level of conflict between parents and adolescents. Interactions with other family members positively or negatively reinforce interactions between parents and their adolescent children. Such interactions between parents and their adolescents maintain the conflict between parents and their adolescent children.

The objectives of this study were to investigate the nature, extent, and intensity of parent–adolescent conflicts during the social movements of 2019, with a special focus on conflicts over their differences in political views. This study also aimed at studying the conflict resolution strategies adopted when adolescents have conflicts with their parents during this period. It also aimed at investigating the relationship of the perceived parenting styles of adolescents of their parents and their problem-solving styles in resolving conflicts with their parents. It is our hypothesis that there was a high intensity of conflicts between adolescents and their parents on differences in political views in this period. It is also our hypothesis that those parents who adopted an authoritative parenting style had a lower intensity of conflicts with their adolescents. As the behavioral systems perspective highlights the important role of the reciprocal responses of parents in conflict episodes, adolescents’ perceived and expected parenting styles as well as whether parenting styles predict adolescent problem-solving strategies were also examined. The study was approved by the Research Ethics Committee of the City University of Hong Kong (REC Ref. no.: H002156).

## 2. Materials and Methods

### 2.1. Respondents and Procedure

After stratification based on geographic location, an invitation letter was sent to 90 randomly selected secondary schools of the total 506 in Hong Kong at that time. Students aged between 11 and 16 years living in Hong Kong in 2019, as well as their parents, were included in the study. Schools forwarded the link to an online questionnaire to students and their parents who met the inclusion criteria between April and September 2020. Students not living in Hong Kong in 2019 or not living with their parents were not included. Incomplete questionnaires were also omitted from analysis. Written consent from responding students and their parents was obtained before the students were invited to complete the questionnaire. After completing the questionnaire, respondents were issued a computer-generated code and given a supermarket coupon worth HKD50. The questionnaire comprised adaptations of various existing scales to incorporate activities in which adolescents participated during the social movements.

### 2.2. Measures

#### 2.2.1. Demographic Data

Adolescents were asked to provide basic information about their age, gender, place of birth, years living in Hong Kong, religious affiliation and activity, number of household members, their parents’ marital status, occupation, education level, and household income.

#### 2.2.2. Conflict Issue Checklist

The Conflict Issue Checklist [22] was included in the second part of the questionnaire. This is a three-point Likert scale that asks respondents to report the frequency and intensity of conflicts with their parents. This is a checklist with 44 items. Sample topics include “doing homework” and “putting away clothes”. It is designed to assess parent–adolescent conflict. “Human chains”, “Lennon walls”, “demonstration”, and “road obstruction” were added to the original list of conflict issues to ensure the inclusion of activities associated with the protests.

#### 2.2.3. Parenting Styles and Dimensions Questionnaire (PSDQ)

The PSDQ [23] investigates respondents’ perceived and expected parenting styles. It measures authoritative, authoritarian, and permissive parenting styles, and comprises 32 items rated on a 5-point Likert scale ranging from 1 (never) to 5 (always). The Chinese version of the PSDQ, tested in a local study, indicated that the alpha coefficients for the three scales are as follows: authoritative parenting (α = 0.85), authoritarian parenting (α = 0.71), and permissive parenting (α = 0.66). The authoritative scale is composed of 15 items that assess parental warmth and involvement, parental reasoning/induction, parents’ encouragement of democratic participation, and parents’ good-natured behaviors. The sample items include “I am responsive to our child’s feelings and needs”. The authoritarian scale includes 12 items that assess parents’ verbal hostility, parental corporal punishment, non-reasoning/punitive parenting strategies, and parents’ directiveness. The sample items include “I shout at him”. Finally, the permissive scale comprises 5 items measuring dimensions of parenting, including a lack of follow through, ignoring misbehavior, and self-confidence. Robinson et al. reported internal reliability coefficients of 0.81 (authoritarian), 0.83 (authoritative), and 0.65 (permissive). The Chinese version of the PSDQ, tested in a local study, indicated the following alpha coefficients for the three scales: authoritative parenting (α = 0.85), authoritarian parenting (α = 0.71), and permissive parenting (α = 0.66). The content validity index (S-CVI) was 0.97 in one study [24]. This scale has been applied in a local study [25] to fit the local context. The examination of its psychometric properties in this study indicates internally consistent scores.

#### 2.2.4. The Ineffective Arguing Scale

The ineffective arguing scale was used to assess how two people handle conflicts with each other [14]. This is an 8-item measure asking respondents to describe how they argue with each other during conflicts. Sample questions include “our arguments were left unresolved” and “we would go for days without settling our differences”. The scale uses a 5-point Likert scale for responses. It ranges from strongly disagree to strongly agree. Higher scores indicate higher levels of ineffective arguing. This scale has been used in research to examine the conflict resolution styles in different types of relationships.

#### 2.2.5. Conflict Resolution Questionnaire

The final part of the questionnaire involved Kurdek’s conflict resolution questionnaire [14], which was originally assigned to measure conflict resolution in couples, but was adapted for the parent–child context. The validity of this adapted measure has been demonstrated in a previous study [26]. The questionnaire measures four conflict resolution styles (5 items each): positive problem solving, conflict engagement, withdrawal, and compliance. The scale measures four conflict resolution styles: positive problem solving, conflict engagement, withdrawal, and compliance. The scale was validated in Hong Kong [26]. Cronbach’s alphas for positive problem solving were 0.83 and 0.81, 0.75, and 0.81 for conflict engagement, 0.72 and 0.70 for withdrawal, and 0.78 and 0.81 for compliance for adolescent and mother reports, respectively.

#### 2.2.6. Data Analysis

The nature, frequency, and intensity of conflicts between respondents and their parents will first be explored. A correlation matrix will be constructed to illustrate the correlations between all conflict issues, differences in political stances, and differences in the expression of political stances with their parents. Then, the actual conflict resolution strategies adopted among adolescents during this period and ideal conflict resolution strategies will be explored and compared with a *t*-test. A regression analysis will then be performed to investigate the relationship between the conflict resolution strategies adopted by adolescents, their perceived parenting styles, and the intensity of conflicts. All data analyses were performed using SPSS version 26 software.

## 3. Results

### 3.1. Demographic Data

Eight hundred and sixty-six valid questionnaires were completed. Table 1 summarizes the respondents’ demographic details. Two hundred and seventy respondents were male and five hundred and ninety-six were female. Their mean age was 14.7 years. Of the respondents, 768 (88.7%) were born in Hong Kong; 667 (77%) had no religion; and 818 (94.4%) reported no chronic illness. Of them, 234 (27%) were only children, 473 (54.6%) had one sibling, and 36 (18.3%) had two siblings or more. Most (666, 76.9%) reported that their parents were married, 89 (10.3%) had divorced parents, and 86 (9.8%) reported that their parents were cohabiting, widowed, or separated. Of the respondents, 709 (81.9%) lived with their father, 791 (91.3%) lived with their mother, and 558 (64.4%) lived with their siblings. Most fathers (657; 75.9%) and 360 (41.6%) mothers were in full-time employment. Only 1 (0.1%) father and 49 (5.7%) mothers were unemployed. Nearly half of the respondents’ fathers (431, 49.7%) completed their education at secondary school, and 215 (24.8%) had post-secondary education; 552 (63.7%) mothers completed their education at secondary school, and 191 (22.1%) had post-secondary education. Nearly two-thirds of respondents (553; 63.9%) reported a household income of HKD 30,000 or less per month, and 127 (14.7%) reported more than HKD 50,000 (the median household income in Hong Kong in 2020 was HKD 25,500) [27]. One hundred and eighty-one (20.9%) families were recipients of Hong Kong’s financial safety net for citizens unable to support themselves financially, the Comprehensive Social Security Assistance (CSSA) scheme (financial assistance from the government). Just over half (450; 51.9%) of the respondents had different political stances from their father, and a similar proportion (444; 51.1%) had different political stances from their mother.

### 3.2. Issue of Conflicts between Parents and Adolescents during the Social Movements in 2019

Pearson’s correlation matrix of demographic data (Table 2) illustrates that there was a significant negative correlation of adolescents’ ages and their differences in political stances with their fathers (r = −0.232, *p* < 0.01) and their mothers (r = −0.254, *p* < 0.01). This indicated that those that are younger have higher levels of differences in political stances with their parents. Differences in political stances are also positively related to adolescents’ fathers’ (r = 0.239, *p* < 0.01) and mothers’ education levels (r = 0.190, *p* < 0.01). This may be due to the higher the education level, the more that parents could discuss their political views with their children, and thus have more differences.

Table 3 identifies the five most frequent causes of conflict with parents reported by respondents: using smartphones (58.9%); bedtime (48.2%); talking back to parents (41.9%); fighting with siblings (38.9%); and cleaning up their bedroom (38.5%). Nearly a quarter (24.1%) reported conflict with their parents over their political views, and 20.4% reported conflict with their parents over expressing political views. Just over a third (36.8%) of the respondents indicated having frequent conflicts about political differences, and 34.5% indicated having frequent conflicts about expressing their political views. Of all the identified causes for conflict, differences in political stances were responsible for the most severe intensity of conflict, reported by 51.2% of the respondents. This was closely followed by ways of expressing political views, with 45.8% of the respondents reporting that this led to severe conflict.

Table 3 also reveals that 38.9% of the respondents encountered conflict with their parents because of fighting with their siblings, 34.4% reporting that these conflicts were severe. One-third (33.5%) reported conflict with their parents for making too much noise at home, and a quarter (25.5%) reported this as high-intensity conflict. A third (33.5%) reported being bothered by parents when they wanted to be alone, three-quarters indicating that such conflicts were of a medium to high intensity.

### 3.3. Correlation of Conflict Issue Checklist

A Pearson’s correlation matrix of the different issues causing conflict, differences in political stances, and differences in ways of expressing political stances was also generated. Table 4 illustrates that conflict because of differences in political stances and expressing political views was significantly positively related to other conflict issues, except doing homework, using phones, and fighting with siblings.

### 3.4. Conflict Resolution Strategies

Table 5 indicates a higher proportion of respondents reported medium to high scores in using conflict engagement strategies (648; 74.8%), withdrawal conflict resolution strategies (486; 56.2%), and compliance conflict resolution strategies (635; 73.4%) during the social movements of 2019 as opposed to their ideal strategies. With regard to respondents’ ideal strategies, most respondents reported wishing to engage in positive problem solving with their parents (848; 97.9%), followed by compliance (782; 90.3%).

With regard to the mean differences in the scores of the four different strategies between the social movements’ strategies and respondents’ ideal strategies, Table 6 indicates higher scores for conflict engagement and withdrawal employed during the social movements than for the ideal strategies. The difference is statistically significant (*p* < 0.001). Higher scores for the ideal strategies than the actual strategies employed during the social movements were observed for positive problem solving and compliance problem solving; the difference was statistically significant (*p* < 0.001).

### 3.5. Perceived Parenting Style

Simple linear regression was used to test if parental parenting styles significantly predicted adolescents’ use of conflict resolution strategies and the intensity of conflicts over differences in political stances and ways of expressing them during social movements. As Table 7 indicates, the overall regression was statistically significant in ineffective arguing (R^2^ = 0.44, F(3) = 232.4, *p* = 0.001), conflict engagement (R^2^ = 0.36, F(3) = 163.5, *p* = 0.001), positive problem-solving (R^2^ = 0.31, F(3) = 128.9, *p* = 0.001), withdrawal (R^2^ = 0.24, F(3) = 93.0, *p* = 0.001), and compliance (R^2^ = 0.30, F(3) = 27.6, *p* = 0.001) conflict resolution strategies.

Authoritarian (B = 0.23, t = 11.43, *p* = 0.001) and permissive parenting styles (B = 0.74, t = 8.6, *p* = 0.001) statistically significantly predicted ineffective arguing, conflict engagement (authoritarian B = −0.16, t = 14.71, *p* = 0.001) (permissive B = −0.26, t = −5.67, *p* = 0.001), and withdrawal (authoritarian B = 0.10, t = 0.32, *p* = 0.001) (permissive B = −0.26, t = −5.67, *p* = 0.001) conflict resolution strategies among adolescents. In addition, authoritarian parenting styles statistically significantly predicted compliance (B = 0.06, t = 6.7, *p* = 0.001). Conversely, authoritarian (B = −0.03, t = −2.27, *p* = 0.05) and permissive parenting styles (B = −0.21, t = −4.05, *p* = 0.001) statistically significantly negatively predicted positive problem solving. 

Authoritative parenting styles statistically significantly predicted positive problem solving (B = 0.29, t = 9.66, *p* = 0.001). Conversely, an authoritarian parenting style statistically significantly predicted ineffective arguing (B = 0.23, t = 11.43, *p* = 0.001), negatively predicted conflict engagement (B = −0.15, t = −5.5, *p* = 0.001), and withdrawal (B = −0.2, t = −6.9, *p* = 0.001) conflict resolution strategies. Nevertheless, neither authoritative parenting nor permissive parenting styles predicted a compliance conflict resolution strategy.

Simple linear regression was also used to test if parental parenting styles predicted conflict intensity over differences in political stances and conflict intensity over differences in the expression of political stances (Table 8). The overall regression was statistically significant for conflict intensity over differences in political stances (R^2^ = 0.08, F(3) = 5.96, *p* = 0.002) and conflict intensity over differences in the expression of political stances (R^2^ = 0.08, F(3) = 5.1, *p* = 0.001). A parental authoritative parenting style significantly negatively predicted conflict intensity over differences in political stances (B = −0.07, t = −2.6, *p* = 0.05). An authoritarian parenting style significantly positively predicted the intensity of conflict over the expression of political stances (B = 0.02, t = 2.1, *p* = 0.05).

## 4. Discussion

This study explored parent–adolescent conflict in Hong Kong, the conflict resolution strategies adopted by adolescents, and the relationship with parents’ parenting styles during the social movements in 2019. During this period of exceptional social crisis, only 24.1% of respondents encountered conflict with their parents because of differences in political stances, and only 20.4% encountered conflict with their parents over expressing political views. Although these were not high proportions, strong correlations were observed with other parent–adolescent conflict issues. This may be because of the high emotional involvement in political differences, and such differences and conflict affect other areas of conflict with parents. This further illustrates that adolescents’ differences with their parents on political stances affected their relationship with their parents in general during the social movements in Hong Kong in 2019. With the high correlation with other sources of conflict, differences in political stances and ways of expressing political views generally reflected the differences in the core values of parents and adolescents. Furthermore, more than half (50.1%) of the respondents reported high conflict intensity when they had different political stances or ways of expressing their political stances to their parents. This is the highest proportion of adolescents expressing angry or very angry emotions during this period. Concerning the correlation of demographic data with differences in political stances with their parents, younger respondents had more conflict over this issue than older respondents. Parents may exert more control over early adolescents than late adolescents. Similar to other studies, the major reason for parent–adolescent conflict was disputes over smartphone use. For the most part, however, this generated relatively low-intensity conflict. Another study finding was that 38.9% of the respondents indicated conflict with their siblings. The Hong Kong Government suspended classes in November 2019 because of escalating social unrest in the territory, and this may have resulted in increasing contact between siblings and, therefore, the scope for intersibling conflict.

Concerning the problem-solving strategies adopted by adolescents, most of the respondents used conflict engagement and withdrawal techniques in conflicts with their parents. Withdrawal techniques include remaining silent for a long period, refusing to talk, or distancing. Nevertheless, when respondents were asked about their ideal strategies for resolving conflict with their parents, most preferred problem-solving strategies involving discussing issues constructively, through negotiation, compromise, and finding alternatives to the problem. Respondents, thus, wished to use more positive than negative measures to resolve conflict with their parents. However, in the highly emotional and conflictual period of the social movements for Hong Kong society as a whole, more adolescents used conflict engagement and withdrawal rather than problem-solving strategies. That means that in an environment of social unrest, the wider system affected parent–child relationships in the family system, especially conflict resolution strategies.

Respondents who perceived their parents’ parenting style as being authoritative employed less ineffective arguing and conflict engagement strategies. Conversely, respondents who perceived their parents’ parenting style as being authoritarian experienced higher levels of conflict engagement with their parents. Authoritative parenting, in which parents express warmth, understanding, and empathy towards their adolescent children, while having reasonable expectations of their behavior, engender more positive conflict resolution strategies in their children. Therefore, authoritative parenting could help resolve parent–adolescent conflict during social movements, especially when adolescents and their parents hold different political views and ways of expressing them.

### 4.1. Limitations of Present Study

This study only uncovered relationships and associations between conflicts of adolescents with their parents and differences in their political stances during the social movements in Hong Kong in 2019. It does not provide any possible causality or direction of the relationship between these variables. The correlation was also not able to reflect the complex nature of the parent–adolescent relationship in which other variables not able to be measured were not covered. The scope of the survey was also limited because of the difficulties in collecting data during COVID-19. It could only reach those adolescents who had parental consent to fill in the questionnaire and was also limited to those who could access an Online Survey through internet. Moreover, this is a retrospective study in which the respondents were reflecting on a period that they had gone through a few months ago. The accuracy of the study depends on the respondents’ ability to accurately recall their experiences. 

### 4.2. Conclusions

This study explored adolescents’ perspectives of parent–adolescent conflict during the social movements in Hong Kong in 2019. It contributes to the research gap on parent–adolescent conflict during this period. It identified the kinds of conflicts between adolescents and parents, as well as conflict resolution strategies. It contributes to the behavioral family system perspective confirming the exacerbation of conflictual relationships when parents adopt an authoritarian parenting style and adolescents adopt conflict engagement and withdrawal strategies, resulting in more parent–adolescent conflict. On the contrary, an authoritative parenting style has been identified as being correlated with positive problem-solving strategies among adolescents in conflict situations. This study also contributes to our understanding that in a time of social unrest, adolescents’ emotions were intensified, leading to more conflict with parents in general. Future research could usefully explore whether Chinese adolescents are inclined to withdraw during conflict with their parents. Further exploration can also be done on other factors that contribute to parent–adolescent conflict, moderation analyses of different conflict resolution strategies, and the negative outcomes of parent–adolescent conflict.

## Figures and Tables

**Table 1 behavsci-13-00756-t001:** Demographic data of respondents.

		(*N* = 866)
		n (%)
Gender	Male	270 (31.2)
	Female	596 (68.8)
Age	Mean (s.d.)	14.7 (1.93)
Form	Mean (s.d.)	3.61 (1.70)
Years of residence in Hong Kong	Mean (s.d.)	13.2 (4.06) (Missing: 18)
Place of birth	Hong Kong	768 (88.7)
	China	92 (10.6)
	Other	6 (0.7)
	India	1 (0.1)
	Macau	2 (0.2)
	England	2 (0.2)
Religious beliefs	No	667 (77.0)
	Yes	199 (23.0)
	Christianity	166 (19.2)
	Buddhism	23 (2.7)
	Islam	2 (0.2)
	Taoism	3 (0.3)
	Other	5 (0.6)
Chronic illness	No	818 (94.4)
	Yes	48 (5.5)
Siblings	None	234 (27)
	1	473 (54.6)
	2	123 (14.2)
	3	29 (3.3)
	More than 3	7 (0.8)
Parents’ marital status	Married	666 (76.9)
	Remarriage	15 (1.7)
	Divorced	89 (10.3)
	Separated	22 (2.5)
	Cohabitation	42 (4.8)
	Widowed	22 (2.5)
	Other	10 (1.2)
	Unknown	1 (0.1)
	Children’s home	1 (0.1)
	Single parent	1 (0.1)
	Unmarried	1 (0.1)
Cohabitation	Father	709 (81.9)
	Mother	791 (91.3)
	Grandparents	134 (15.4)
	Relatives	56 (6.6)
	Siblings	558 (64.4)
	Other	31 (3.6)
	Domestic helper	13 (1.5)
	Children’s home	3 (0.3)
	Stepfather	1 (0.1)
	Live alone	2 (0.2)
	Best friend	1 (0.1)
Employment status (father)	Full time	657 (75.9)
	Part time (~22 h)	43 (5.0)
	Daily/hourly paid (<22 h)	30 (3.5)
	Housework	16 (1.8)
	Unemployed	59 (6.8)
	Other	61 (7.0)
	Unknown/not sure	13 (1.5)
	N/A (divorced/single parent)	11 (1.3)
	Daily wage (>40 h)	1 (0.1)
	Retired	6 (0.7)
	Self-employed	1 (0.1)
	Unemployed (temporary)	1 (0.1)
Educational level (father)	No formal education	12 (1.4)
	Primary school	61 (7.0)
	Form 1 to 3	180 (20.8)
	Form 4 to 5	190 (21.9)
	Form 6 to 7	154 (17.8)
	Technical institute/vocational training	54 (6.2)
	College level or above	215 (24.8)
Employment status (mother)	Full time	360 (41.6)
	Part time (~22 h/wk)	77 (8.9)
	Daily/hourly paid	50 (5.8)
	Housework	308 (35.6)
	Unemployed	49 (5.7)
	Other	22 (2.5)
	Unknown/not sure	4 (0.5)
	N/A	2 (0.2)
	Self-employed	1 (0.1)
	Freelance	1 (0.1)
	Unemployed	1 (0.1)
Educational level (mother)	No formal education	11 (1.3)
	Primary school	79 (9.1)
	Form 1 to 3	176 (20.3)
	Form 4 to 5	205 (23.7)
	Form 6 to 7	171 (19.7)
	Technical institute/vocational training	33 (3.8)
	College level or above	191 (22.1)
Monthly household income (HKD)	10,000 or below	121 (14.0)
	10,001 to 30,000	432 (49.9)
	30,001 to 50,000	186 (21.5)
	50,001 or above	127 (14.7)
CSSA (family living on financial assistance from the government)	Yes	181 (20.9)
	No	685 (79.1)
Political stance (father)	Very dissimilar	123 (14.2)
	Dissimilar	98 (11.3)
	Slightly dissimilar	229 (26.4)
	Similar	293 (33.8)
	Very similar	123 (14.2)
Political stance (mother)	Very dissimilar	92 (10.5)
	Dissimilar	113 (13.0)
	Slightly dissimilar	239 (27.6)
	Similar	286 (33.0)
	Very similar	136 (15.7)

**Table 2 behavsci-13-00756-t002:** Bivariate correlations of demographic data and differences in political stances with father and mother.

	Father Political Stances	Mother Political Stances
Gender	−0.039	−0.043
Age	−0.232 **	−0.254 **
Form	−0.242 **	−0.280 **
Years of residence in Hong Kong	−0.184 **	−0.187 **
Place of birth	0.010	−0.021
Religious beliefs	0.048	0.056
Religion involvement	0.016	0.052
Chronic illness	−0.049	−0.033
Number of siblings	−0.010	−0.037
Parents’ marital status	−0.096 **	−0.041
Father employment status	−0.113 **	−0.031
Father education level	0.239 **	0.190 **
Mother employment status	0.000	0.008
Mother education level	0.148 **	0.139 **
Monthly income	0.079 *	0.052
CSSA (family living on financial assistance from the government)	0.091 **	0.054

* *p* < 0.05; ** *p* < 0.01.

**Table 3 behavsci-13-00756-t003:** Nature and intensity of conflicts with parents during the social movements in 2019.

Nature of Conflict	Total ^a^	Frequency ^b^			Intensity ^c^			
		Low	Medium	High	Low	Medium	High	Mean
Difference in political stance	209 (24.1)	74 (35.4)	58 (27.8)	77 (36.8)	34 (16.3)	68 (32.5)	107 (51.2)	3.6
Ways of expressing political views	177 (20.4)	72 (40.7)	44 (24.9)	61 (34.5)	39 (22)	57 (32.2)	81 (45.8)	3.4
Fighting with siblings	337 (38.9)	108 (32)	101 (30)	128 (38)	63 (18.7)	158 (46.9)	116 (34.4)	3.2
Talking back to parents	363 (41.9)	150 (41.3)	96 (26.4)	117 (32.2)	84 (23.1)	181 (49.9)	98 (27)	3.1
Lying	160 (18.5)	86 (54.1)	45 (28.3)	28 (17.6)	58 (36.3)	48 (30)	54 (33.8)	3.1
Bothering parents when they want to be left alone	31 (3.6)	13 (43.3)	5 (16.7)	12 (40)	12 (38.7)	14 (45.2)	5 (16.1)	3.0
Using smartphones	510 (58.9)	229 (44.9)	133 (26.1)	148 (29)	130 (25.5)	280 (54.9)	100 (19.6)	2.9
Making too much noise at home	290 (33.5)	132 (45.7)	76 (26.3)	81 (28)	97 (33.4)	118 (40.7)	75 (25.9)	2.9
Doing homework	273 (31.5)	112 (41.5)	71 (26.3)	87 (21.3)	93 (34.1)	121 (44.3)	59 (21.6)	2.9
Messing up the house	177 (20.4)	72 (40.7)	45 (25.4)	60 (33.9)	56 (31.6)	78 (44.1)	43 (24.3)	2.9
Getting poor grades at school	272 (31.4)	143 (53.2)	65 (24.2)	61 (22.6)	115 (42.3)	93 (34.2)	64 (23.5)	2.8
Computer use	218 (25.2)	89 (41.6)	61 (28.5)	64 (29.9)	75 (34.4)	98 (45)	45 (20.6)	2.8
Coming home on time	196 (22.6)	98 (50)	55 (28.1)	43 (21.9)	88 (44.9)	64 (32.7)	44 (22.4)	2.8
Choice of friends	99 (11.4)	61 (62.2)	28 (28.6)	9 (9.2)	42 (42.4)	31 (31.3)	26 (26.2)	2.8
Bedtime	417 (48.2)	177 (42.5)	96 (23.1)	143 (34.3)	174 (41.7)	169 (40.5)	74 (17.7)	2.7
Cleaning up bedroom	333 (38.5)	161 (48.6)	93 (28.1)	77 (23.3)	141 (42.3)	144 (43.2)	48 (14.4)	2.7
How to spend free time	249 (28.8)	78 (31.5)	70 (28.2)	100 (40.3)	104 (41.8)	97 (39)	48 (19.3)	2.7
Cursing	151 (17.4)	86 (57)	25 (16.6)	40 (26.5)	70 (46.4)	48 (31.8)	33 (21.9)	2.7
Taking care of CDs, games, bikes, pets, and things	116 (13.4)	53 (45.7)	33 (28.4)	30 (25.9)	55 (47.4)	42 (36.2)	19 (16.4)	2.7
Getting in trouble at school	54 (6.2)	35 (64.8)	13 (24.1)	6 (11.2)	24 (44.4)	17 (31.5)	13 (24.1)	2.7
Going on dates	50 (5.8)	29 (58)	13 (26)	8 (16)	18 (36)	22 (44)	10 (20)	2.7
Helping out around the house	265 (30.6)	129 (49)	69 (26.2)	65 (24.7)	124 (46.8)	104 (39.2)	37 (14)	2.6
Being bothered by parents when wanting to be left alone	251 (29)	98 (39)	69 (27.5)	84 (33.5)	61 (24.3)	126 (50.2)	64 (25.5)	2.6
Getting up in the morning	194 (22.4)	59 (30.4)	57 (29.4)	78 (40.2)	96 (49.5)	70 (36.1)	28 (14.4)	2.6
Putting away clothes	259 (29.9)	137 (53.3)	58 (22.6)	62 (24.1)	132 (51)	98 (37.8)	29 (11.2)	2.5
How money is spent	195 (22.5)	105 (54.1)	54 (27.8)	35 (18)	105 (53.8)	56 (28.7)	34 (17.4)	2.5
What time to have meals	141 (16.3)	64 (45.4)	34 (24.1)	43 (30.5)	66 (46.8)	57 (40.4)	18 (12.7)	2.5
Going out with parents (shopping, going to cinemas)	134 (15.5)	75 (56)	42 (31.3)	17 (12.7)	68 (50.7)	46 (34.3)	20 (14.9)	2.5
Playing music or radio too loudly	132 (15.2)	64 (48.9)	48 (36.6)	19 (14.6)	66 (50)	54 (40.9)	12 (9.1)	2.5
Getting to school on time	81 (9.4)	40 (49.4)	22 (27.2)	19 (23.5)	42 (51.9)	28 (34.6)	11 (13.5)	2.5
Cleanliness (washing, showering, and brushing teeth)	189 (21.8)	90 (47.6)	44 (23.3)	55 (29.1)	103 (54.5)	66 (34.9)	20 (10.6)	2.4
What to wear	132 (15.2)	81 (61.8)	26 (19.8)	24 (18.4)	72 (54.5)	48 (36.4)	12 (9.1)	2.4
Allowance	123 (14.2)	67 (55.4)	32 (26.4)	22 (18.1)	71 (57.7)	31 (25.2)	21 (17.1)	2.4
What teenager eats	111 (12.8)	35 (31.5)	30 (27)	46 (41.4)	67 (60.4)	34 (30.6)	10 (9)	2.4
Buying CDs, games, toys, and things	78 (9)	55 (70.5)	12 (15.4)	11 (14.1)	41 (52.6)	31 (39.7)	6 (7.7)	2.4
Selecting new clothes	94 (10.9)	63 (67)	18 (19.1)	13 (13.8)	55 (58.5)	32 (34)	7 (7.4)	2.3
Picking books or movies	41 (4.7)	29 (70.7)	7 (17.1)	5 (12.2)	23 (56.1)	14 (34.1)	4 (9.8)	2.3
Clothing neatness	66 (7.6)	42 (64.6)	12 (18.5)	11 (16.9)	46 (69.7)	14 (21.2)	6 (9.1)	2.1

*N* = 866. ^a^ Total number of respondents indicating having this conflict with their parents. ^b^ Number of times respondents have conflict with their parents per week. High: 7–10 times; medium: 4–6 times; and low: 1–3 times. ^c^ Intensity level: High: angry; very angry. Medium: moderately angry. Low: calm; very calm.

**Table 4 behavsci-13-00756-t004:** Correlation matrix of differences in political stances and differences in expressing political views, with conflict issues between adolescents and parents.

Conflict Issues	Differences in Political Stances with Parents	Differences in Expressing Political Views with Parents
1. Using phones	0.066	0.037
2. Time for going to bed	0.166 **	0.090 **
3. Cleaning up bedroom	0.169 **	0.177 **
4. Doing homework	0.027	0.045
5. Putting away clothes	0.160 **	0.140 **
6. Using computer	0.146 **	0.151 **
7. Cleanliness (washing, showering, and brushing teeth)	0.076 *	0.116 **
8. What to wear	0.099 **	0.135 **
9. Clothing neatness	0.116 **	0.144 **
10. Making too much noise at home	0.159 **	0.183 **
11. Fighting with siblings	0.060	0.028
12. Cursing	0.191 **	0.209 **
13. How money is spent	0.263 **	0.271 **
14. Picking books or movies	0.173 **	0.199 **
15. Allowance	0.223 **	0.262 **
16. Going out with parents (shopping, going to cinemas)	0.105 **	0.162 **
17. Playing music too loudly	0.247 **	0.199 **
18. Taking care of CDs, games, bikes, and pets	0.111 **	0.124 **
19. Drinking beer or other alcohol	0.214 **	0.270 **
20. Buying CDs, games, toys, and things	0.185 **	0.247 **
21. Going on dates	0.201 **	0.288 **
22. Choice of friends	0.072 *	0.116 **
23. Selecting new clothes	0.175 **	0.243 **
24. Coming home on time	0.218 **	0.240 **
25. Getting to school on time	0.127 **	0.198 **
26. Getting poor grades at school	0.103 **	0.126 **
27. Getting into trouble at school	0.169 **	0.237 **
28. Lying	0.111 **	0.141 **
29. Helping out around the house	0.210 **	0.192 **
30. Talking back to parents	0.220 **	0.203 **
31. Getting up in the morning	0.206 **	0.172 **
32. Bothering parents when they want to be left alone	0.177 **	0.264 **
33. Bothering teenagers when they want to be left alone	0.243 **	0.231 **
34. Messing up the house	0.194 **	0.205 **
35. Time to have meals	0.205 **	0.200 **
36. How to spend free time	0.151 **	0.137 **
37. Earning money away from the house	0.241 **	0.289 **
38. What teenager eats	0.160 **	0.180 **

* *p* < 0.05; ** *p* < 0.001.

**Table 5 behavsci-13-00756-t005:** Number of respondents reporting the use of conflict resolution strategies during the social movements in 2019 and their ideal strategies.

	Score of Using the Selected Strategy					
	Actual Strategy			Ideal Strategy		
	Low	Medium	High	Low	Medium	High
Conflict engagement	218 (25.2%)	577 (66.6%)	71 (8.2%)	461 (53.2%)	376 (43.4%)	29 (3.3%)
Positive problem solving	40 (4.6%)	483 (55.8%)	343 (39.6%)	18 (2.1%)	171 (19.7%)	677 (78.2%)
Withdrawal	230 (26.6%)	450 (52.0%)	36 (4.2%)	429 (49.5%)	404 (46.7%)	33 (3.8%)
Compliance	231 (26.7%)	592 (68.4%)	43 (5.0%)	84 (9.7%)	735 (84.9%)	47 (5.4%)

*N* = 866.

**Table 6 behavsci-13-00756-t006:** Conflict resolution strategies during the social movements in 2019 and the ideal situation.

	Social Movements in 2019		Ideal Situation			
	M	SD	M	SD	t	*p*
Conflict engagement	8.5	2.9	6.9	2.6	15.2	0.000
Positive problem solving	11.8	3.2	15.1	3.2	−28.1	0.000
Withdrawal	7.4	2.8	5.8	2.5	17.0	0.000
Compliance	6.8	2.2	7.9	1.8	−12.8	0.000

*N* = 866.

**Table 7 behavsci-13-00756-t007:** Regression analysis between parenting styles and conflict resolution strategies.

	R^2^	F	B	β	t
Ineffective arguing scale	0.44	232.4			
Parental authoritative style			−0.65	−0.96	−12.51 **
Parental authoritarian style			0.23	0.35	11.43 **
Parental permissive style			0.74	0.66	8.60 **
Conflict engagement	0.36	163.5			
Parental authoritative style	.		−0.15	−0.46	−5.50 **
Parental authoritarian style			0.16	0.48	14.71 **
Parental permissive style			0.26	0.47	5.67 **
Positive problem solving	0.31	128.9			
Parental authoritative style			0.29	0.83	9.66 **
Parental authoritarian style			−0.03	−0.08	−2.27 *
Parental permissive style			−0.21	−0.35	−4.05 **
Withdrawal	0.24	93.0			
Parental authoritative style			−0.20	−0.62	−6.9 **
Parental authoritarian style			0.10	0.32	8.9 **
Parental permissive style			0.27	0.50	5.6 **
Compliance	0.30	27.6			
Parental authoritative style			−0.03	−0.14	−1.38
Parental authoritarian style			0.06	0.26	6.7 **
Parental permissive style			0.06	0.15	1.48

* *p* < 0.05; ** *p* < 0.001.

**Table 8 behavsci-13-00756-t008:** Regression analysis between parenting styles and conflict intensity over differences in political stances.

	R^2^	F	B	β	t
Conflict intensity over differences in political stances	0.08	5.96			
Parental authoritative style			−0.07	−0.50	−2.65 *
Parental authoritarian style			0.01	0.09	1.2
Parental permissive style			0.08	0.35	1.9
Conflict intensity over differences in expressing political stances	0.08	5.1			
Parental authoritative style			−0.05	−0.35	−1.8
Parental authoritarian style			0.02	1.8	2.1 *
Parental permissive style			0.08	0.33	1.74

* *p* < 0.05.

## Data Availability

The data that support the findings of this study are available from the corresponding author upon reasonable request.

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
