# Peer review of "Parent–Adolescent Conflict: Adolescents’ Coping Strategies and Preferred Parenting Styles during the Social Movements in Hong Kong, 2019"

_behavsci, 2023, doi:10.3390/bs13090756_

Round 1
Reviewer 1 Report
This study examined an important issue. However, this manuscript needs major revisions.
Introduction
1. “Hong Kong adolescents have also reported conflicts with parents because of smoking and choice of leisure activities.” Please add the reference.
2. The aims and hypotheses of this study needed revision. Clear aims and hypotheses of study can lead the statistical analysis and result in reasonable results.
Methods
1. I believe that the location of Table 1 in the text warrants revision. The MDPI template often results in dislocation of tables. Table 1 should be moved into Results. Moreover, Table 1 seemed to cover some contents of Methods.
2. The content of Table 1 also needs revision. For example, Employment Status Father; Educational Level (father); Employment Status Mother; Educational Level (mother); CSSA; missing parts after CSSA…
3. Please describe statistical analysis plans in Methods.
Results
1. Because this study lacked clear study aims and statistical analysis plans, the contents of Results were not easy to follow. for example, it was hard to understand what “Conflict resolution strategies during the social movement 2019 and ideal situation (Table 5)” meant.
2. “Table 3 illustrates that conflict because of differences in political stance and expressing political views was significantly positively related to other conflict issues except doing homework and fighting with siblings.” Their association with using phones were also nonsignificant.
3. “This indicates that adolescents who conflicted with their parents on differences… social movements in Hong Kong in 2019.” This paragraph should be moved to Discussion.
4. Was it the aim of this study to examine the associations between parenting style and conflict resolution strategies? It did not relate to parent-adolescent conflicts.
5. This study collected a lot of demographic characteristics. However, the authors did not examine the roles of demographic characteristics in parent-adolescent conflicts.
Discussion
1. “Younger respondents had more conflict over this issue than older respondents.” I did not find this result in any table.
2. “With the high correlation with other sources of conflict, differences in political stance and ways of expressing political views generally encouraged other parent-adolescent conflict.” How did the author make this assumption? It is more possible that both parent-adolescent conflicts for daily issues and political stance came from the differences in the core values.
3. The contents of discussion need major revisions to connect with the aims and hypotheses of this study.
4. This study had many limitations. The author did not pay attention to them.
This study examined an important issue. However, this manuscript needs major revisions.
Reviewer 2 Report
Thank you for providing me with the opportunity to review this manuscript. The topic presented in the paper is interesting, as is aimed examined the Parent-adolescent conflict: Adolescents’ coping strategies and preferred parenting style during the social movements in Hong
Kong 2019.
The strengths of the study are the timely and accurate technical description, as well as the graphic and tabular layout and also the high number of participants. However, the work has some aspects that should be better attended to by the authors.
1) The introduction should be revised in translation, with a more synthesis, some passages are not smooth and there is too much redundancy and repetition.
2) The description of the instruments is not accurate also needs, in my opinion, to be expanded.
3) the discussion should be revised in traslation
4) conclusions need to be expanded
5) Rewrite and subdivide the abstract according journal standards
6)The limitations of the study are lacking
Consult and add the following studies:
- https://www.ijpsy.com/volumen15/num2/415/factor-structure-and-criterion-validity-EN.pdf
- Dentale, F., Verrastro, V., Petruccelli, I., Diotaiuti, P., Petruccelli, F., Cappelli, L., & Martini, P. S. (2015). Relationship between parental narcissism and children’s mental vulnerability: Mediation role of rearing style. International Journal of Psychology & Psychological Therapy, 15(3), 337–347.
- Verrastro, V., Petruccelli, I., Diotaiuti, P., Petruccelli, F., Dentale, F., & Barbaranelli, C. (2016). Self-Serving Bias in the Implicit and Explicit Evaluation of Partners and Exes as Parents: A Pilot Study. Psychological reports, 118(1), 251–265. https://doi.org/10.1177/0033294115626819
Reviewer 3 Report
- I have read with attention; I think there is a real good link between general purpose and results.
Review: The authors defined a good theoretical framework about Nature and extent of parent–adolescent conflict and about Moderating role of conflict resolution strategies and parent–adolescent conflict.
Reviewer 4 Report
Dear authors: First of all, congratulate you for such an interesting manuscript or research carried out on the conflicts between parents and children in the field of political thought or political issues, developed in the social unrest of Hong Kong in 2019. However, I see that necessary to improve some aspects of the manuscript.
1. In table 1, they would need to comment on what the CSSA acronym means, within the table itself. It is true that when you comment on the results of that table, they mention what those acronyms mean. However, I see it appropriate that it be transcribed in the table.
2. In the Conflict Issue Checklist, I would like you to indicate the number of items that correspond to the scale.
3. In PSDQ, it would be appropriate to include the number of items that correspond to each dimension or parenting style.
4. Regarding the different scales or instruments used, I see it as necessary that they include not only Cronbach's Alpha but other reliability and validity indices such as McDonald's Omega, composite reliability and the average variance extracted.
5. I do not see the treatment that they have carried out with the lost cases.
6. They should include a section with the methodology used, that is, the statistical analyzes carried out, whether correlations, independent samples t tests or linear regressions, in addition to commenting on the statistical programs or packages used to obtain the results and carry out statistical analyses.
Round 2
Reviewer 1 Report
The authors have revised their manuscript based on the reviewer's suggestions. I would like to suggest the editors accepting it for publication.
Author Response
Thanks to your comments
Reviewer 2 Report
The authors have made all changes enriching the work in all its sections. Now the work has a dignity of publication. I thank the authors for executing my directions in a precise and timely manner.
Author Response
Thanks for your comments and advise.
Reviewer 4 Report
Dear authors: I have checked the changes made and they seem correct. The only thing, a mistake in the value of a correlation coefficient r, which is on line 243.
Author Response
Dear Reviewer,
Thank you for your comments. I have revised the error on line 243 accrodingly. Enclosed please find the revised file.
